# Adaptive Online Learning in Dynamic Environments

**Lijun Zhang,    Shiyin Lu,    Zhi-Hua Zhou**
National Key Laboratory for Novel Software Technology
Nanjing University, Nanjing 210023, China
{zhanglj, lusy, zhouzh}@lamda.nju.edu.cn

## Abstract

In this paper, we study online convex optimization in dynamic environments, and aim to bound the dynamic regret with respect to *any* sequence of comparators. Existing work have shown that online gradient descent enjoys an $O(\sqrt{T}(1 + P_T))$ dynamic regret, where $T$ is the number of iterations and $P_T$ is the path-length of the comparator sequence. However, this result is unsatisfactory, as there exists a large gap from the $\Omega(\sqrt{T(1 + P_T)})$ lower bound established in our paper. To address this limitation, we develop a novel online method, namely adaptive learning for dynamic environment (Ader), which achieves an optimal $O(\sqrt{T(1 + P_T)})$ dynamic regret. The basic idea is to maintain a set of experts, each attaining an optimal dynamic regret for a specific path-length, and combines them with an expert-tracking algorithm. Furthermore, we propose an improved Ader based on the surrogate loss, and in this way the number of gradient evaluations per round is reduced from $O(\log T)$ to 1. Finally, we extend Ader to the setting that a sequence of dynamical models is available to characterize the comparators.

## 1   Introduction

Online convex optimization (OCO) has become a popular learning framework for modeling various real-world problems, such as online routing, ad selection for search engines and spam filtering [Hazan, 2016]. The protocol of OCO is as follows: At iteration $t$, the online learner chooses $\mathbf{x}_t$ from a convex set $\mathcal{X}$. After the learner has committed to this choice, a convex cost function $f_t : \mathcal{X} \mapsto \mathbb{R}$ is revealed. Then, the learner suffers an instantaneous loss $f_t(\mathbf{x}_t)$, and the goal is to minimize the cumulative loss over $T$ iterations. The standard performance measure of OCO is regret:

$$\sum_{t=1}^{T} f_t(\mathbf{x}_t) - \min_{\mathbf{x} \in \mathcal{X}} \sum_{t=1}^{T} f_t(\mathbf{x}) \tag{1}$$

which is the cumulative loss of the learner minus that of the best constant point chosen in hindsight.

The notion of regret has been extensively studied, and there exist plenty of algorithms and theories for minimizing regret [Shalev-Shwartz et al., 2007, Hazan et al., 2007, Srebro et al., 2010, Duchi et al., 2011, Shalev-Shwartz, 2011, Zhang et al., 2013]. However, when the environment is changing, the traditional regret is no longer a suitable measure, since it compares the learner against a *static* point. To address this limitation, recent advances in online learning have introduced an enhanced measure—*dynamic* regret, which received considerable research interest over the years [Hall and Willett, 2013, Jadbabaie et al., 2015, Mokhtari et al., 2016, Yang et al., 2016, Zhang et al., 2017].

In the literature, there are two different forms of dynamic regret. The general one is introduced by Zinkevich [2003], who proposes to compare the cumulative loss of the learner against *any* sequence

of comparators

$$R(\mathbf{u}_1, \ldots, \mathbf{u}_T) = \sum_{t=1}^{T} f_t(\mathbf{x}_t) - \sum_{t=1}^{T} f_t(\mathbf{u}_t) \tag{2}$$

where $\mathbf{u}_1, \ldots, \mathbf{u}_T \in \mathcal{X}$. Instead of following the definition in (2), most of existing studies on dynamic regret consider a restricted form, in which the sequence of comparators consists of local minimizers of online functions [Besbes et al., 2015], i.e.,

$$R(\mathbf{x}_1^*, \ldots, \mathbf{x}_T^*) = \sum_{t=1}^{T} f_t(\mathbf{x}_t) - \sum_{t=1}^{T} f_t(\mathbf{x}_t^*) = \sum_{t=1}^{T} f_t(\mathbf{x}_t) - \sum_{t=1}^{T} \min_{\mathbf{x} \in \mathcal{X}} f_t(\mathbf{x}) \tag{3}$$

where $\mathbf{x}_t^* \in \operatorname{argmin}_{\mathbf{x} \in \mathcal{X}} f_t(\mathbf{x})$ is a minimizer of $f_t(\cdot)$ over domain $\mathcal{X}$. Note that although $R(\mathbf{u}_1, \ldots, \mathbf{u}_T) \leq R(\mathbf{x}_1^*, \ldots, \mathbf{x}_T^*)$, it does not mean the notion of $R(\mathbf{x}_1^*, \ldots, \mathbf{x}_T^*)$ is stronger, because an upper bound for $R(\mathbf{x}_1^*, \ldots, \mathbf{x}_T^*)$ could be very loose for $R(\mathbf{u}_1, \ldots, \mathbf{u}_T)$.

The general dynamic regret in (2) includes the static regret in (1) and the restricted dynamic regret in (3) as special cases. Thus, minimizing the general dynamic regret can automatically adapt to the nature of environments, stationary or dynamic. In contrast, the restricted dynamic regret is too pessimistic, and unsuitable for stationary problems. For example, it is meaningless to the problem of statistical machine learning, where $f_t$'s are sampled independently from the *same* distribution. Due to the random perturbation caused by sampling, the local minimizers could differ significantly from the global minimizer of the expected loss. In this case, minimizing (3) will lead to overfitting.

Because of its flexibility, we focus on the general dynamic regret in this paper. Bounding the general dynamic regret is very challenging, because we need to establish a *universal* guarantee that holds for any sequence of comparators. By comparison, when bounding the restricted dynamic regret, we only need to focus on the local minimizers. Till now, we have very limited knowledge on the general dynamic regret. One result is given by Zinkevich [2003], who demonstrates that online gradient descent (OGD) achieves the following dynamic regret bound

$$R(\mathbf{u}_1, \ldots, \mathbf{u}_T) = O\left(\sqrt{T}(1 + P_T)\right) \tag{4}$$

where $P_T$, defined in (5), is the path-length of $\mathbf{u}_1, \ldots, \mathbf{u}_T$.

However, the linear dependence on $P_T$ in (4) is too loose, and there is a large gap between the upper bound and the $\Omega(\sqrt{T(1 + P_T)})$ lower bound established in our paper. To address this limitation, we propose a novel online method, namely adaptive learning for dynamic environment (Ader), which attains an $O(\sqrt{T(1 + P_T)})$ dynamic regret. Ader follows the framework of learning with expert advice [Cesa-Bianchi and Lugosi, 2006], and is inspired by the strategy of maintaining multiple learning rates in MetaGrad [van Erven and Koolen, 2016]. The basic idea is to run multiple OGD algorithms in parallel, each with a different step size that is optimal for a specific path-length, and combine them with an expert-tracking algorithm. While the basic version of Ader needs to query the gradient $O(\log T)$ times in each round, we develop an improved version based on surrogate loss and reduce the number of gradient evaluations to 1. Finally, we provide extensions of Ader to the case that a sequence of dynamical models is given, and obtain tighter bounds when the comparator sequence follows the dynamical models closely.

The contributions of this paper are summarized below.

- We establish the *first* lower bound for the general regret bound in (2), which is $\Omega(\sqrt{T(1 + P_T)})$.
- We develop a serial of novel methods for minimizing the general dynamic regret, and prove an optimal $O(\sqrt{T(1 + P_T)})$ upper bound.
- Compared to existing work for the restricted dynamic regret in (3), our result is *universal* in the sense that the regret bound holds for any sequence of comparators.
- Our result is also *adaptive* because the upper bound depends on the path-length of the comparator sequence, so it automatically becomes small when comparators change slowly.

## 2  Related Work

In this section, we provide a brief review of related work in online convex optimization.

## 2.1 Static Regret

In static setting, online gradient descent (OGD) achieves an $O(\sqrt{T})$ regret bound for general convex functions. If the online functions have additional curvature properties, then faster rates are attainable. For strongly convex functions, the regret bound of OGD becomes $O(\log T)$ [Shalev-Shwartz et al., 2007]. The $O(\sqrt{T})$ and $O(\log T)$ regret bounds, for convex and strongly convex functions respectively, are known to be minimax optimal [Abernethy et al., 2008]. For exponentially concave functions, Online Newton Step (ONS) enjoys an $O(d \log T)$ regret, where $d$ is the dimensionality [Hazan et al., 2007]. When the online functions are both smooth and convex, the regret bound could also be improved if the cumulative loss of the optimal prediction is small [Srebro et al., 2010].

## 2.2 Dynamic Regret

To the best of our knowledge, there are only two studies that investigate the general dynamic regret [Zinkevich, 2003, Hall and Willett, 2013]. While it is impossible to achieve a sublinear dynamic regret in general, we can bound the dynamic regret in terms of certain regularity of the comparator sequence or the function sequence. Zinkevich [2003] introduces the path-length

$$P_T(\mathbf{u}_1, \ldots, \mathbf{u}_T) = \sum_{t=2}^{T} \|\mathbf{u}_t - \mathbf{u}_{t-1}\|_2 \tag{5}$$

and provides an upper bound for OGD in (4). In a subsequent work, Hall and Willett [2013] propose a variant of path-length

$$P_T'(\mathbf{u}_1, \ldots, \mathbf{u}_T) = \sum_{t=1}^{T} \|\mathbf{u}_{t+1} - \Phi_t(\mathbf{u}_t)\|_2 \tag{6}$$

in which a sequence of dynamical models $\Phi_t(\cdot) : \mathcal{X} \mapsto \mathcal{X}$ is incorporated. Then, they develop a new method, dynamic mirror descent, which achieves an $O(\sqrt{T}(1 + P_T'))$ dynamic regret. When the comparator sequence follows the dynamical models closely, $P_T'$ could be much smaller than $P_T$, and thus the upper bound of Hall and Willett [2013] could be tighter than that of Zinkevich [2003].

For the restricted dynamic regret, a powerful baseline, which simply plays the minimizer of previous round, i.e., $\mathbf{x}_{t+1} = \operatorname{argmin}_{\mathbf{x} \in \mathcal{X}} f_t(\mathbf{x})$, attains an $O(P_T^*)$ dynamic regret [Yang et al., 2016], where

$$P_T^* := P_T(\mathbf{x}_1^*, \ldots, \mathbf{x}_T^*) = \sum_{t=2}^{T} \|\mathbf{x}_t^* - \mathbf{x}_{t-1}^*\|_2.$$

OGD also achieves the $O(P_T^*)$ dynamic regret, when the online functions are strongly convex and smooth [Mokhtari et al., 2016], or when they are convex and smooth and all the minimizers lie in the interior of $\mathcal{X}$ [Yang et al., 2016]. Another regularity of the comparator sequence is the squared path-length

$$S_T^* := S_T(\mathbf{x}_1^*, \ldots, \mathbf{x}_T^*) = \sum_{t=2}^{T} \|\mathbf{x}_t^* - \mathbf{x}_{t-1}^*\|_2^2$$

which could be smaller than the path-length $P_T^*$ when local minimizers move slowly. Zhang et al. [2017] propose online multiple gradient descent, and establish an $O(\min(P_T^*, S_T^*))$ regret bound for (semi-)strongly convex and smooth functions.

In a recent work, Besbes et al. [2015] introduce the functional variation

$$F_T := F(f_1, \ldots, f_T) = \sum_{t=2}^{T} \max_{\mathbf{x} \in \mathcal{X}} |f_t(\mathbf{x}) - f_{t-1}(\mathbf{x})|$$

to measure the complexity of the function sequence. Under the assumption that an upper bound $V_T \geq F_T$ is known beforehand, Besbes et al. [2015] develop a restarted online gradient descent, and prove its dynamic regret is upper bounded by $O(T^{2/3}(V_T + 1)^{1/3})$ and $O(\log T \sqrt{T(V_T + 1)})$ for convex functions and strongly convex functions, respectively. One limitation of this work is that the bounds are not adaptive because they depend on the upper bound $V_T$. So, even when the actual functional variation $F_T$ is small, the regret bounds do not become better.

One regularity that involves the gradient of functions is

$$D_T = \sum_{t=1}^{T} \|\nabla f_t(\mathbf{x}_t) - \mathbf{m}_t\|_2^2$$

where $\mathbf{m}_1, \dots, \mathbf{m}_T$ is a predictable sequence computable by the learner [Chiang et al., 2012, Rakhlin and Sridharan, 2013]. From the above discussions, we observe that there are different types of regularities. As shown by Jadbabaie et al. [2015], these regularities reflect distinct aspects of the online problem, and are not comparable in general. To take advantage of the smaller regularity, Jadbabaie et al. [2015] develop an adaptive method whose dynamic regret is on the order of $\sqrt{D_T + 1} + \min\{\sqrt{(D_T + 1)P_T^*}, (D_T + 1)^{1/3}T^{1/3}F_T^{1/3}\}$. However, it relies on the assumption that the learner can calculate each regularity online.

## 2.3 Adaptive Regret

Another way to deal with changing environments is to minimize the adaptive regret, which is defined as maximum static regret over any contiguous time interval [Hazan and Seshadhri, 2007]. For convex functions and exponentially concave functions, Hazan and Seshadhri [2007] have developed efficient algorithms that achieve $O(\sqrt{T \log^3 T})$ and $O(d \log^2 T)$ adaptive regrets, respectively. Later, the adaptive regret of convex functions is improved [Daniely et al., 2015, Jun et al., 2017]. The relation between adaptive regret and restricted dynamic regret is investigated by Zhang et al. [2018b].

# 3 Our Methods

We first state assumptions about the online problem, then provide our motivations, including a lower bound of the general dynamic regret, and finally present the proposed methods as well as their theoretical guarantees. All the proofs can be found in the full paper [Zhang et al., 2018a].

## 3.1 Assumptions

Similar to previous studies in online learning, we introduce the following common assumptions.

**Assumption 1** *On domain $\mathcal{X}$, the values of all functions belong to the range $[a, a + c]$, i.e.,*
$$a \leq f_t(\mathbf{x}) \leq a + c, \ \forall \mathbf{x} \in \mathcal{X}, \ and \ t \in [T].$$

**Assumption 2** *The gradients of all functions are bounded by $G$, i.e.,*
$$\max_{\mathbf{x} \in \mathcal{X}} \|\nabla f_t(\mathbf{x})\|_2 \leq G, \ \forall t \in [T]. \tag{7}$$

**Assumption 3** *The domain $\mathcal{X}$ contains the origin $\mathbf{0}$, and its diameter is bounded by $D$, i.e.,*
$$\max_{\mathbf{x}, \mathbf{x}' \in \mathcal{X}} \|\mathbf{x} - \mathbf{x}'\|_2 \leq D. \tag{8}$$

Note that Assumptions 2 and 3 imply Assumption 1 with any $c \geq GD$. In the following, we assume the values of $G$ and $D$ are known to the leaner.

## 3.2 Motivations

According to Theorem 2 of Zinkevich [2003], we have the following dynamic regret bound for online gradient descent (OGD) with a constant step size.

**Theorem 1** *Consider the online gradient descent (OGD) with $\mathbf{x}_1 \in \mathcal{X}$ and*
$$\mathbf{x}_{t+1} = \Pi_{\mathcal{X}}\big[\mathbf{x}_t - \eta \nabla f_t(\mathbf{x}_t)\big], \ \forall t \geq 1$$
*where $\Pi_{\mathcal{X}}[\cdot]$ denotes the projection onto the nearest point in $\mathcal{X}$. Under Assumptions 2 and 3, OGD satisfies*
$$\sum_{t=1}^{T} f_t(\mathbf{x}_t) - \sum_{t=1}^{T} f_t(\mathbf{u}_t) \leq \frac{7D^2}{4\eta} + \frac{D}{\eta} \sum_{t=2}^{T} \|\mathbf{u}_{t-1} - \mathbf{u}_t\|_2 + \frac{\eta T G^2}{2}$$
*for any comparator sequence $\mathbf{u}_1, \dots, \mathbf{u}_T \in \mathcal{X}$.*

Thus, by choosing $\eta = O(1/\sqrt{T})$, OGD achieves an $O(\sqrt{T}(1 + P_T))$ dynamic regret, that is universal. However, this upper bound is far from the $\Omega(\sqrt{T(1 + P_T)})$ lower bound indicated by the theorem below.

**Theorem 2** *For any online algorithm and any $\tau \in [0, TD]$, there exists a sequence of comparators $\mathbf{u}_1, \ldots, \mathbf{u}_T$ satisfying Assumption 3 and a sequence of functions $f_1, \ldots, f_T$ satisfying Assumption 2, such that*
$$P_T(\mathbf{u}_1, \ldots, \mathbf{u}_T) \leq \tau \text{ and } R(\mathbf{u}_1, \ldots, \mathbf{u}_T) = \Omega\big(G\sqrt{T(D^2 + D\tau)}\big).$$

Although there exist lower bounds for the restricted dynamic regret [Besbes et al., 2015, Yang et al., 2016], to the best of our knowledge, this is the *first* lower bound for the general dynamic regret.

Let's drop the universal property for the moment, and suppose we only want to compare against a *specific* sequence $\bar{\mathbf{u}}_1, \ldots, \bar{\mathbf{u}}_T \in \mathcal{X}$ whose path-length $\overline{P}_T = \sum_{t=2}^{T} \|\bar{\mathbf{u}}_t - \bar{\mathbf{u}}_{t-1}\|_2$ is known beforehand. In this simple setting, we can tune the step size optimally as $\eta^* = O(\sqrt{(1 + \overline{P}_T)/T})$ and obtain an improved $O(\sqrt{T(1 + \overline{P}_T)})$ dynamic regret bound, which matches the lower bound in Theorem 2. Thus, when bounding the general dynamic regret, we face the following challenge: On one hand, we want the regret bound to hold for *any* sequence of comparators, but on the other hand, to get a tighter bound, we need to tune the step size for a *specific* path-length. In the next section, we address this dilemma by running multiple OGD algorithms with different step sizes, and combining them through a meta-algorithm.

### 3.3 The Basic Approach

Our proposed method, named as adaptive learning for dynamic environment (Ader), is inspired by a recent work for online learning with multiple types of functions—MetaGrad [van Erven and Koolen, 2016]. Ader maintains a set of experts, each attaining an optimal dynamic regret for a different path-length, and chooses the best one using an expert-tracking algorithm.

**Meta-algorithm**     Tracking the best expert is a well-studied problem [Herbster and Warmuth, 1998], and our meta-algorithm, summarized in Algorithm 1, is built upon the exponentially weighted average forecaster [Cesa-Bianchi and Lugosi, 2006]. The inputs of the meta-algorithm are its own step size $\alpha$, and a set $\mathcal{H}$ of step sizes for experts. In Step 1, we active a set of experts $\{E^\eta | \eta \in \mathcal{H}\}$ by invoking the expert-algorithm for each $\eta \in \mathcal{H}$. In Step 2, we set the initial weight of each expert. Let $\eta_i$ be the $i$-th smallest step size in $\mathcal{H}$. The weight of $E^{\eta_i}$ is chosen as

$$w_1^{\eta_i} = \frac{C}{i(i+1)}, \text{ and } C = 1 + \frac{1}{|\mathcal{H}|}. \tag{9}$$

In each round, the meta-algorithm receives a set of predictions $\{\mathbf{x}_t^\eta | \eta \in \mathcal{H}\}$ from all experts (Step 4), and outputs the weighted average (Step 5):

$$\mathbf{x}_t = \sum_{\eta \in \mathcal{H}} w_t^\eta \mathbf{x}_t^\eta$$

where $w_t^\eta$ is the weight assigned to expert $E^\eta$. After observing the loss function, the weights of experts are updated according to the exponential weighting scheme (Step 7):

$$w_{t+1}^\eta = \frac{w_t^\eta e^{-\alpha f_t(\mathbf{x}_t^\eta)}}{\sum_{\mu \in \mathcal{H}} w_t^\mu e^{-\alpha f_t(\mathbf{x}_t^\mu)}}.$$

In the last step, we send the gradient $\nabla f_t(\mathbf{x}_t^\eta)$ to each expert $E^\eta$ so that they can update their own predictions.

**Expert-algorithm**     Experts are themselves algorithms, and our expert-algorithm, presented in Algorithm 2, is the standard online gradient descent (OGD). Each expert is an instance of OGD, and takes the step size $\eta$ as its input. In Step 3 of Algorithm 2, each expert submits its prediction $\mathbf{x}_t^\eta$ to the meta-algorithm, and receives the gradient $\nabla f_t(\mathbf{x}_t^\eta)$ in Step 4. Then, in Step 5 it performs gradient descent

$$\mathbf{x}_{t+1}^\eta = \Pi_\mathcal{X}\big[\mathbf{x}_t^\eta - \eta \nabla f_t(\mathbf{x}_t^\eta)\big]$$

---

**Algorithm 1** Ader: Meta-algorithm

---

**Require:** A step size $\alpha$, and a set $\mathcal{H}$ containing step sizes for experts
1: Activate a set of experts $\{E^\eta | \eta \in \mathcal{H}\}$ by invoking Algorithm 2 for each step size $\eta \in \mathcal{H}$
2: Sort step sizes in ascending order $\eta_1 \leq \eta_2 \leq \cdots \leq \eta_N$, and set $w_1^{\eta_i} = \frac{C}{i(i+1)}$
3: **for** $t = 1, \ldots, T$ **do**
4:     Receive $\mathbf{x}_t^\eta$ from each expert $E^\eta$
5:     Output
$$\mathbf{x}_t = \sum_{\eta \in \mathcal{H}} w_t^\eta \mathbf{x}_t^\eta$$
6:     Observe the loss function $f_t(\cdot)$
7:     Update the weight of each expert by
$$w_{t+1}^\eta = \frac{w_t^\eta e^{-\alpha f_t(\mathbf{x}_t^\eta)}}{\sum_{\mu \in \mathcal{H}} w_t^\mu e^{-\alpha f_t(\mathbf{x}_t^\mu)}}$$
8:     Send gradient $\nabla f_t(\mathbf{x}_t^\eta)$ to each expert $E^\eta$
9: **end for**

---

**Algorithm 2** Ader: Expert-algorithm

---

**Require:** The step size $\eta$
1: Let $\mathbf{x}_1^\eta$ be any point in $\mathcal{X}$
2: **for** $t = 1, \ldots, T$ **do**
3:     Submit $\mathbf{x}_t^\eta$ to the meta-algorithm
4:     Receive gradient $\nabla f_t(\mathbf{x}_t^\eta)$ from the meta-algorithm
5:
$$\mathbf{x}_{t+1}^\eta = \Pi_{\mathcal{X}} \left[ \mathbf{x}_t^\eta - \eta \nabla f_t(\mathbf{x}_t^\eta) \right]$$
6: **end for**

---

to get the prediction for the next round.

Next, we specify the parameter setting and our dynamic regret. The set $\mathcal{H}$ is constructed in the way such that for any possible sequence of comparators, there exists a step size that is nearly optimal. To control the size of $\mathcal{H}$, we use a geometric series with ratio 2. The value of $\alpha$ is tuned such that the upper bound is minimized. Specifically, we have the following theorem.

**Theorem 3** *Set*
$$\mathcal{H} = \left\{ \eta_i = \frac{2^{i-1}D}{G}\sqrt{\frac{7}{2T}} \,\middle|\, i = 1, \ldots, N \right\} \tag{10}$$
*where $N = \lceil \frac{1}{2}\log_2(1 + 4T/7) \rceil + 1$, and $\alpha = \sqrt{8/(Tc^2)}$ in Algorithm 1. Under Assumptions 1, 2 and 3, for* any *comparator sequence $\mathbf{u}_1, \ldots, \mathbf{u}_T \in \mathcal{X}$, our proposed Ader method satisfies*
$$\sum_{t=1}^{T} f_t(\mathbf{x}_t) - \sum_{t=1}^{T} f_t(\mathbf{u}_t) \leq \frac{3G}{4}\sqrt{2T(7D^2 + 4DP_T)} + \frac{c\sqrt{2T}}{4}\left[1 + 2\ln(k+1)\right]$$
$$= O\left(\sqrt{T(1 + P_T)}\right)$$
*where*
$$k = \left\lfloor \frac{1}{2}\log_2\left(1 + \frac{4P_T}{7D}\right) \right\rfloor + 1. \tag{11}$$

The order of the upper bound matches the $\Omega(\sqrt{T(1 + P_T)})$ lower bound in Theorem 2 exactly.

## 3.4 An Improved Approach

The basic approach in Section 3.3 is simple, but it has an obvious limitation: From Steps 7 and 8 in Algorithm 1, we observe that the meta-algorithm needs to query the value and gradient of $f_t(\cdot)$

$N$ times in each round, where $N = O(\log T)$. In contrast, existing algorithms for minimizing static regret, such as OGD, only query the gradient *once* per iteration. When the function is complex, the evaluation of gradients or values could be expensive, and it is appealing to reduce the number of queries in each round.

**Surrogate Loss** We introduce *surrogate loss* [van Erven and Koolen, 2016] to replace the original loss function. From the first-order condition of convexity [Boyd and Vandenberghe, 2004], we have

$$f_t(\mathbf{x}) \geq f_t(\mathbf{x}_t) + \langle \nabla f_t(\mathbf{x}_t), \mathbf{x} - \mathbf{x}_t \rangle, \ \forall \mathbf{x} \in \mathcal{X}.$$

Then, we define the surrogate loss in the $t$-th iteration as

$$\ell_t(\mathbf{x}) = \langle \nabla f_t(\mathbf{x}_t), \mathbf{x} - \mathbf{x}_t \rangle \tag{12}$$

and use it to update the prediction. Because

$$f_t(\mathbf{x}_t) - f_t(\mathbf{u}_t) \leq \ell_t(\mathbf{x}_t) - \ell_t(\mathbf{u}_t), \tag{13}$$

we conclude that the regret w.r.t. true losses $f_t$'s is smaller than that w.r.t. surrogate losses $\ell_t$'s. Thus, it is safe to replace $f_t$ with $\ell_t$. The new method, named as improved Ader, is summarized in Algorithms 3 and 4.

**Meta-algorithm** The new meta-algorithm in Algorithm 3 differs from the old one in Algorithm 1 since Step 6. The new algorithm queries the gradient of $f_t(\cdot)$ at $\mathbf{x}_t$, and then constructs the surrogate loss $\ell_t(\cdot)$ in (12), which is used in subsequent steps. In Step 8, the weights of experts are updated based on $\ell_t(\cdot)$, i.e.,

$$w_{t+1}^\eta = \frac{w_t^\eta e^{-\alpha \ell_t(\mathbf{x}_t^\eta)}}{\sum_{\mu \in \mathcal{H}} w_t^\mu e^{-\alpha \ell_t(\mathbf{x}_t^\mu)}}.$$

In Step 9, the gradient of $\ell_t(\cdot)$ at $\mathbf{x}_t^\eta$ is sent to each expert $E^\eta$. Because the surrogate loss is linear,

$$\nabla \ell_t(\mathbf{x}_t^\eta) = \nabla f_t(\mathbf{x}_t), \ \forall \eta \in \mathcal{H}.$$

As a result, we only need to send the same $\nabla f_t(\mathbf{x}_t)$ to all experts. From the above descriptions, it is clear that the new algorithm only queries the gradient once in each iteration.

**Expert-algorithm** The new expert-algorithm in Algorithm 4 is almost the same as the previous one in Algorithm 2. The only difference is that in Step 4, the expert receives the gradient $\nabla f_t(\mathbf{x}_t)$, and uses it to perform gradient descent

$$\mathbf{x}_{t+1}^\eta = \Pi_{\mathcal{X}} \left[ \mathbf{x}_t^\eta - \eta \nabla f_t(\mathbf{x}_t) \right]$$

in Step 5.

We have the following theorem to bound the dynamic regret of the improved Ader.

**Theorem 4** *Use the construction of $\mathcal{H}$ in (10), and set $\alpha = \sqrt{2/(TG^2 D^2)}$ in Algorithm 3. Under Assumptions 2 and 3, for* any *comparator sequence $\mathbf{u}_1, \ldots, \mathbf{u}_T \in \mathcal{X}$, our improved Ader method satisfies*

$$\sum_{t=1}^T f_t(\mathbf{x}_t) - \sum_{t=1}^T f_t(\mathbf{u}_t) \leq \frac{3G}{4} \sqrt{2T(7D^2 + 4DP_T)} + \frac{GD\sqrt{2T}}{2} \left[ 1 + 2\ln(k+1) \right]$$

$$= O\left( \sqrt{T(1 + P_T)} \right)$$

*where $k$ is defined in (11).*

Similar to the basic approach, the improved Ader also achieves an $O(\sqrt{T(1 + P_T)})$ dynamic regret, that is universal and adaptive. The main advantage is that the improved Ader only needs to query the gradient of the online function *once* in each iteration.

---

**Algorithm 3** Improved Ader: Meta-algorithm

---

**Require:** A step size $\alpha$, and a set $\mathcal{H}$ containing step sizes for experts
1: Activate a set of experts $\{E^\eta | \eta \in \mathcal{H}\}$ by invoking Algorithm 4 for each step size $\eta \in \mathcal{H}$
2: Sort step sizes in ascending order $\eta_1 \leq \eta_2 \leq \cdots \leq \eta_N$, and set $w_1^{\eta_i} = \frac{C}{i(i+1)}$
3: **for** $t = 1, \ldots, T$ **do**
4:    Receive $\mathbf{x}_t^\eta$ from each expert $E^\eta$
5:    Output

$$\mathbf{x}_t = \sum_{\eta \in \mathcal{H}} w_t^\eta \mathbf{x}_t^\eta$$

6:    Query the gradient of $f_t(\cdot)$ at $\mathbf{x}_t$
7:    Construct the surrogate loss $\ell_t(\cdot)$ in (12)
8:    Update the weight of each expert by

$$w_{t+1}^\eta = \frac{w_t^\eta e^{-\alpha \ell_t(\mathbf{x}_t^\eta)}}{\sum_{\mu \in \mathcal{H}} w_t^\mu e^{-\alpha \ell_t(\mathbf{x}_t^\mu)}}$$

9:    Send gradient $\nabla f_t(\mathbf{x}_t)$ to each expert $E^\eta$
10: **end for**

---

---

**Algorithm 4** Improved Ader: Expert-algorithm

---

**Require:** The step size $\eta$
1: Let $\mathbf{x}_1^\eta$ be any point in $\mathcal{X}$
2: **for** $t = 1, \ldots, T$ **do**
3:    Submit $\mathbf{x}_t^\eta$ to the meta-algorithm
4:    Receive gradient $\nabla f_t(\mathbf{x}_t)$ from the meta-algorithm
5:

$$\mathbf{x}_{t+1}^\eta = \Pi_\mathcal{X}\left[\mathbf{x}_t^\eta - \eta \nabla f_t(\mathbf{x}_t)\right]$$

6: **end for**

---

### 3.5 Extensions

Following Hall and Willett [2013], we consider the case that the learner is given a sequence of dynamical models $\Phi_t(\cdot) : \mathcal{X} \mapsto \mathcal{X}$, which can be used to characterize the comparators we are interested in. Similar to Hall and Willett [2013], we assume each $\Phi_t(\cdot)$ is a contraction mapping.

**Assumption 4** *All the dynamical models are contraction mappings, i.e.,*

$$\|\Phi_t(\mathbf{x}) - \Phi_t(\mathbf{x}')\|_2 \leq \|\mathbf{x} - \mathbf{x}'\|_2, \tag{14}$$

*for all $t \in [T]$, and $\mathbf{x}, \mathbf{x}' \in \mathcal{X}$.*

Then, we choose $P_T'$ in (6) as the regularity of a comparator sequence, which measures how much it deviates from the given dynamics.

**Algorithms** For brevity, we only discuss how to incorporate the dynamical models into the basic Ader in Section 3.3, and the extension to the improved version can be done in the same way. In fact, we only need to modify the expert-algorithm, and the updated one is provided in Algorithm 5. To utilize the dynamical model, after performing gradient descent, i.e.,

$$\bar{\mathbf{x}}_{t+1}^\eta = \Pi_\mathcal{X}\left[\mathbf{x}_t^\eta - \eta \nabla f_t(\mathbf{x}_t^\eta)\right]$$

in Step 5, we apply the dynamical model to the intermediate solution $\bar{\mathbf{x}}_{t+1}^\eta$, i.e.,

$$\mathbf{x}_{t+1}^\eta = \Phi_t(\bar{\mathbf{x}}_{t+1}^\eta),$$

and obtain the prediction for the next round. In the meta-algorithm (Algorithm 1), we only need to replace Algorithm 2 in Step 1 with Algorithm 5, and the rest is the same. The dynamic regret of the new algorithm is given below.

---
**Algorithm 5** Ader: Expert-algorithm with dynamical models
---
**Require:** The step size $\eta$, a sequence of dynamical models $\Phi_t(\cdot)$
1: Let $\mathbf{x}_1^\eta$ be any point in $\mathcal{X}$
2: **for** $t = 1, \ldots, T$ **do**
3:    Submit $\mathbf{x}_t^\eta$ to the meta-algorithm
4:    Receive gradient $\nabla f_t(\mathbf{x}_t^\eta)$ from the meta-algorithm
5:

$$\bar{\mathbf{x}}_{t+1}^\eta = \Pi_{\mathcal{X}} \left[ \mathbf{x}_t^\eta - \eta \nabla f_t(\mathbf{x}_t^\eta) \right]$$

6:

$$\mathbf{x}_{t+1}^\eta = \Phi_t(\bar{\mathbf{x}}_{t+1}^\eta)$$

7: **end for**
---

**Theorem 5** *Set*

$$\mathcal{H} = \left\{ \eta_i = \frac{2^{i-1} D}{G} \sqrt{\frac{1}{T}} \,\middle|\, i = 1, \ldots, N \right\} \tag{15}$$

*where $N = \left\lceil \frac{1}{2} \log_2(1 + 2T) \right\rceil + 1$, $\alpha = \sqrt{8/(Tc^2)}$, and use Algorithm 5 as the expert-algorithm in Algorithm 1. Under Assumptions 1, 2, 3 and 4, for* any *comparator sequence $\mathbf{u}_1, \ldots, \mathbf{u}_T \in \mathcal{X}$, our proposed Ader method satisfies*

$$\sum_{t=1}^{T} f_t(\mathbf{x}_t) - \sum_{t=1}^{T} f_t(\mathbf{u}_t) \le \frac{3G}{2} \sqrt{T(D^2 + 2DP_T')} + \frac{c\sqrt{2T}}{4} \left[ 1 + 2\ln(k+1) \right]$$

$$= O\left( \sqrt{T(1 + P_T')} \right)$$

*where*

$$k = \left\lfloor \frac{1}{2} \log_2 \left( 1 + \frac{2P_T'}{D} \right) \right\rfloor + 1.$$

Theorem 5 indicates our method achieves an $O(\sqrt{T(1 + P_T')})$ dynamic regret, improving the $O(\sqrt{T}(1 + P_T'))$ dynamic regret of Hall and Willett [2013] significantly. Note that when $\Phi_t(\cdot)$ is the identity map, we recover the result in Theorem 3. Thus, the upper bound in Theorem 5 is also optimal.

## 4   Conclusion and Future Work

In this paper, we study the general form of dynamic regret, which compares the cumulative loss of the online learner against an arbitrary sequence of comparators. To this end, we develop a novel method, named as adaptive learning for dynamic environment (Ader). Theoretical analysis shows that Ader achieves an optimal $O(\sqrt{T(1 + P_T)})$ dynamic regret. When a sequence of dynamical models is available, we extend Ader to incorporate this additional information, and obtain an $O(\sqrt{T(1 + P_T')})$ dynamic regret.

In the future, we will investigate whether the curvature of functions, such as strong convexity and smoothness, can be utilized to improve the dynamic regret bound. We note that in the setting of the restricted dynamic regret, the curvature of functions indeed makes the upper bound tighter [Mokhtari et al., 2016, Zhang et al., 2017]. But whether it improves the general dynamic regret remains an open problem.

### Acknowledgments

This work was partially supported by the National Key R&D Program of China (2018YFB1004300), NSFC (61603177), JiangsuSF (BK20160658), YESS (2017QNRC001), and Microsoft Research Asia.

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
