[Reviews · NeurIPS 2018]

Reviewer 1



This paper studies online convex optimization in dynamic environments, where one wishes to control the regret with respect to sequences of comparators, as opposed to a static comparator. The complexity of a comparator sequence is measured by its path length P_T, and the goal is to obtain an optimal regret bound simultaneously for all path lengths. A first bound in this setting was established in the pioneering paper [1], which showed that online gradient descent (OGD, projected on the convex compact domain) with step-size 1/sqrt(T) achieves an at most T^{1/2} (1 + P_T) regret for all comparator sequences. However, there is a gap between this upper bound and the (T (1+P_T) )^{1/2} lower bound on worst-case regret established in Theorem 2 of the present paper, which is the first lower bound for this problem. On the other hand, if the path length P_T one wishes to compare against is known in advance, optimally tuning the step-size of OGD with respect to P_T in the OGD regret bound yields optimal (T (1+P_T) )^{1/2} regret for this path length. Building on this observation, two algorithms are introduced which satisfy (T (1+P_T))^{1/2} regret simultaneously for all path lengths P_T, matching the above lower bound, up to a doubly logarithmic term in T. The first algorithm, called Ader, consists of a master which aggregates O(log T) instances of OGD with step-sizes in a geometric grid, using an expert algorithm (namely exponential weights). The standard expert regret bound implies a regret bound with respect to all step-sizes, and hence adaptivity to P_T (Theorem 3). As mentioned in the paper, the idea of aggregating different instances of OGD on a grid of step-sizes was considered in [2], to obtain different kinds of adaptivity guarantees. Note that Ader requires to query O(log T) values and gradients at each step, while OGD only needs 1 gradient. To address this, the authors introduce a variant of their algorithm, called "improved Ader", which consists of Ader run on linearized surrogate losses based on the gradient at the point chosen by the master algorithm (hence, only this gradient is used). Using the standard "gradient trick" (regret with respect to the original loss functions is bounded by regret with respect to the linearized losses) and the regret bound for Ader, a similar optimal (T (1+P_T))^{1/2} regret bound is obtained for improved Ader (Theorem 3). Finally, the algorithm and results are adapted to the setting when one measures the complexity of the sequence in terms of deviation with respect to a dynamical model rather than path length, introduced in [3]. This paper is superbly written and easy to follow; in particular, the "Related Work" section provides a clear and detailed account of existing work on the subject. The lower bound and adaptive upper bound for this problem are novel. The approach is natural and the techniques used somewhat straightforward, which I would argue is a plus rather than a minus. Note: The upper and lower bounds are almost matching, up to a log log T factor. While this additional factor can be safely ignored as almost constant, it turns out that it can be avoided altogether, as follows. Instead of putting a uniform prior on the exponential grid of step-sizes, set a mass 1/(i(i+1)) to the i-th step-size (with i as in Eq. (9)), and set the expert learning rate to alpha = 1/T^{1/2}. The regret bound for the exponential weights wrt i is of order T^{1/2} log (i(i+1)) [more sophisticated experts algorithms would yield (T log (i(i+1)))^{1/2} regret wrt to i, but this does not bring any improvement here], which after substituting for the optimal i^* = O(log P_T) for a given value of P_T, gives a total O( (T (1+P_T))^{1/2} + T^{1/2} log log P_T ) = O ((T (1+P_T))^{1/2}) universal dynamic regret bound. A few minor corrections: - Line 113: "an" --> "a"; - Line 142: "holds" --> "to hold"; - Line 171: "setting" --> "set"; - It may be worth adding, probably around or after Eq (10) and (11), that regret wrt true losses f_t is lower than regret wrt surrogate losses \ell_t, with a line that simply shows it (e.g., f_t (x_t) - f_t (u_t) \leq \ell_t (x_t) - \ell_t (u_t) from (10) and (11)). Although, the current formulation is already clear and the trick is simple and standard enough; - Line 202: a reference to the relevant section and theorem can be added. [1] M. Zinkevich. Online convex programming and generalized infinitesimal gradient ascent. In Proceedings of the 20th International Conference on Machine Learning (ICML), pages 928–936, 2003. [2] T. van Erven and W. M. Koolen. Metagrad: Multiple learning rates in online learning. In Advances in Neural Information Processing Systems 29, pages 3666–3674, 2016. [3] E. C. Hall and R. M. Willett. Dynamical models and tracking regret in online convex programming. In Proceedings of the 30th International Conference on Machine Learning, pages 579–587, 2013.

Reviewer 2



This paper addresses the minimization of general dynamic regret in online convex optimization. The paper provides a lower bound and an improved matching upper bound for the problem depending on the path length of the reference sequence of comparators. The paper also gives an efficient version of the algorithm that only requires 1 computation of the gradient based on a surrogate loss function. The paper is clearly written. The contribution is of interest. However, the approach in the paper seems to be heavily inspired by the work on MetaGrad by van Erven and Koolen and overall does not look very novel or surprising. line 38 'doesnot' -> does not line 113 'an lower bound' -> a lower bound

Reviewer 3



The authors study the online convex optimization problem in a dynamic environment. The objective of the paper is to bound the general dynamic regret, i.e. the regret against any sequence of comparators. In this paper, the first lower bound on the general dynamic regret in terms of path-length $P_T$ and time horizon $T$ is proven. They further develop two algorithms that match the regret lower bound up to a $O(sqrt(loglog(T)/P_T))$ factor. The first algorithm called ‘Ader’ uses O(log(T)) function and gradient evaluations each step. The second algorithm called ‘Improved Ader’ provides significant improvement with one function and gradient evaluations each step. The proposed algorithms rely on a key principle that if $P_T$ is known then online gradient descent (Zinkevich 2013) with a specific choice of step size can attain the optimal general dynamic regret. As $P_T$ is unknown, running separate online gradient descent instances each with its own step size and later combining them through the Hedge algorithm provides the upper bound. Finally, the authors extend their result to a setting where an approximate dynamical model for the comparator sequence is known. Similar upper and lower bounds hold in this setting. Pros: * This work effectively narrows down the gap of achievable regret in online convex optimization problem under dynamic environment upto a $O(sqrt loglog(T) )$ factor in terms of path length $P_T$. This improves the gap by $sqrt P_T$. * The writing is fluent and easy to follow. The ideas are well presented in the main body and the proofs presented in the supplementary part is elegant. Cons: * It is not specified clearly that the proposed algorithms assume the knowledge of domain diameter $D$ and the gradient upper bound $G$.